# Perceptions of Family Health Strategy Professionals About People with Mental Disorders

**DOI:** 10.3390/healthcare13030243

**Published:** 2025-01-25

**Authors:** Carla Aparecida Arena Ventura, Camila Kaori Hayashi, Bruna Sordi Carrara, Igor de Oliveira Reis, Bruno Henrique Longo da Silva, Raquel Helena Hernandez Fernandes, Marciana Fernandes Moll

**Affiliations:** 1Department of Psychiatric Nursing and Human Sciences, Nursing School of Ribeirão Preto, University of São Paulo, São Paulo 14040-902, Brazil; bsordi.c@usp.br (B.S.C.); igordeoliveirareis@usp.br (I.d.O.R.); bruno.longohs@gmail.com (B.H.L.d.S.); raquelhhfernandes@usp.br (R.H.H.F.); 2Faculty of Nursing, State University of Campinas, São Paulo 13083-970, Brazil; c259425@dac.unicamp.br (C.K.H.); marcfmol@unicamp.br (M.F.M.)

**Keywords:** mental disorders, primary health care, social stigma

## Abstract

**Background:** The work of professionals in the Family Health Strategy, which is a part of primary health care in Brazil, is essential for the inclusion and support of people with mental disorders. These professionals’ perceptions of individuals with psychiatric diagnoses can directly influence the quality and effectiveness of the care provided. Therefore, the objective of this study was to explore and understand the perceptions of professionals who work in the Family Health Strategy about people with mental disorders. **Methods**: A qualitative study was carried out through individual interviews with different health professionals from July to September 2021 in a municipality in the state of São Paulo, Brazil. Data were analyzed using thematic analysis. **Results**: Findings showed the existence of a concept of people with mental disorders based on the biomedical paradigm, and they experienced limitations arising from the disease that caused restrictions of opportunities, even though they had the capacity to live socially. **Conclusions**: It is necessary to invest in educational interventions to work on the existing stigma among primary health care professionals.

## 1. Introduction

Mental disorders have been recognized as a significant public health issue and can affect anyone at different stages of life. According to the World Health Organization [1], it is estimated that around 970 million people worldwide live with some form of mental disorder (one in every eight), with 82% of them residing in low- and middle-income countries. In addition to common mental disorders such as anxiety and depression, Brazil diagnoses 75,000 new cases of schizophrenia annually, representing 50 cases per 100,000 inhabitants. In the same country, the incidence of mood disorders ranges from 0.5% [2].

Thus, it is evident that mental disorders are recurrent in Brazil and worldwide, and they tend to predispose individuals to suicide, especially when linked to contemporary psychosocial issues (unemployment, economic and political crises), which have been exacerbated since 2020 with the COVID-19 pandemic [3]. In addition to the increase in situations of mental distress and psychiatric diagnoses during the pandemic period, there was intensification and worsening of symptoms in people who already had an established clinical diagnosis, particularly of affective disorders and schizophrenia [4,5].

This reality highlights the need to improve the provision of mental health care across all levels of health care (primary, secondary, and tertiary). However, significant barriers challenge both the quality of care and accessibility, such as prejudice and discrimination among health care professionals [6,7]. Another critical issue is the fragmentation of health care services, where a lack of communication and integration between the different levels of care and specialties can result in uncoordinated and ineffective care. Bureaucracy and the complexity of health care systems also act as barriers, making it difficult to access continuous and integrated mental health treatments [8].

Considering that primary health care (PHC) represents the organizing axis of Brazilian health care and that it should be the “gateway” to access the services of the Brazilian Unified Health System, the Family Health Strategy (FHS) was structured [9]. The FHS, through the Family Health Units (FHUs), seeks to provide longitudinal care in each area of coverage, which increases access to comprehensive and proactive interdisciplinary care for the most vulnerable and marginalized people, including people with mental disorders [10,11].

The FHS has provided better access to quality care compared to traditional health centers [11]. Moreover, the FHS also serves as a catalyst for changes in the institutional and medicalized approaches for treating individuals with mental disorders, as it is a level of health care that links the challenges of Psychiatric Reform with the accountability of professionals working in this context [12]. However, in the face of the barriers surrounding mental health care in the primary care setting, FHS professionals often feel insecure, powerless, distressed, unprepared, and lacking in knowledge [13].

The Comprehensive Mental Health Action Plan 2013–2030, adopted by the 66th World Health Assembly, establishes clear guidelines for promoting mental health globally [1]. Among its main objectives is the provision of integrated mental health services in community settings, emphasizing the importance of the work of Brazilian health professionals in the FHS. These professionals are at the forefront of community care and play a crucial role in the support and inclusion of people with mental disorders. Their perceptions of individuals with psychiatric diagnoses can also directly influence the quality of care provided [14].

It is noteworthy that this is a theme that needs to be further explored in the Brazilian scenario, which can contribute to primary care professionals to develop actions for mental health through effective multidisciplinary monitoring, since they establish greater proximity to users and their families, which favors the recognition of secondary support network(s) and the history of people’s lives and this makes this level of health care ideal to motivate reflections and changes among health professionals. This reality is expressed in the fact that FSH’s multidisciplinary mental health care focuses on referrals to specialized services, which represents a reflection of the prejudice of these professionals who view all people with mental disorders as beings who are not capable of monitoring at this level. It is also noteworthy that, in the Brazilian organizational system, community health agents have been providing more care centered on patients with mental disorders, when compared to doctors and nurses who work at FSH [15]. Therefore, the objective of this study was to explore and understand the perceptions of professionals who work in the Family Health Strategy about people with mental disorders.

## 2. Materials and Methods

### 2.1. Design and Scenery

This investigation is a part of a cross-sectional project that aimed to investigate the presence of stigma towards people with mental disorders and substance use-related problems in PHC in Brazil. This project was implemented in FHUs, where continuous care is provided with a focus on the promotion, protection, and recovery of health, delivered by a multidisciplinary team, with the FHS playing a central role. The present study refers to the exploratory phase of the project, in which qualitative data were collected through semi-structured individual interviews with FHS health professionals from six FHUs in a municipality in the state of São Paulo, Brazil.

### 2.2. Participants

This research included health professionals who are a part of the FHS team. Each FHS team consists of a nurse, a nursing assistant, a doctor, and four to six full-time community health workers, with each of these professionals performing specific functions and responsibilities according to national guidelines [11]. The study included all professionals from the participating Family Health Units (USF) who had direct contact with users with mental disorders, while interns and residents present at the USFs during data collection were excluded, as they are health professionals working in the service for a limited time.

A total of 82 professionals participated in the research, 82.9% were female, and most of them were between 36 and 57 years old (58.9%). More than half declared that they had completed college education (59.8%). Regarding education, there was a prevalence of middle-level courses, such as nursing assistant (3.7%), nursing technician (7.3%), and oral health assistant (4.9%) and courses like radiology, HR management, and others (12.2%). Still, there was a prevalence of training in higher education courses that do not include those who comprise the Family Health Team (ESF) (38.9%). This is due to community health agents (CHAs) and those professionals who have other education besides the one required in the FHU.

According to the professionals’ statements, the following occupations predominated, i.e., 53.7% CHAs, 8.5% family health doctors, 6.1% nurses, 6.1% dentists, and 6.1 oral health assistants. The majority (42.5%) declared that they had been working at the FHU between 1 and 5 years and 8.5% between 16 and 20 years.

### 2.3. Data Collection

Data collection was conducted between July and September 2021 in reserved rooms at the FHUs, during the professionals’ work breaks. The interviews were conducted by two doctoral students in psychiatric nursing, who have consolidated experience in the field and in qualitative research, under the supervision of a full professor from the university to which they are affiliated. A pre-designed script, developed by the research team and reviewed by mental health professionals, was followed.

The semi-structured interview method was chosen because it requires the creation of a preliminary script with questions based on the guiding theories of the investigation while also allowing for possible questions that may arise during the interview. This enabled the interviewers to add new questions when necessary, aiming to delve deeper into aspects deemed important for the study’s objectives [16].

To characterize the participants, the approach began with a sociodemographic questionnaire, with the central idea being to explore what FHS professionals think about people with mental disorders. Questions like “When I mention people with mental disorders, what do you imagine? What do you think makes you feel this way? What do you think life is like for people with mental disorders? How do you think people with mental disorders feel about life?” were posed to the health professionals. The interviews lasted an average of 40 min, were recorded, and later transcribed verbatim to ensure quality control and accuracy of the content.

### 2.4. Data Analysis

Thematic analysis was used as it allows for a better understanding of discourse by analyzing a social phenomenon. This approach facilitates the synthesis of key elements that constitute communication and the construction of representations, thus enabling better assimilation of the data during the analysis process. To understand the discursive context, six steps were applied and are as follows: (1) data familiarization (the thorough reading of the discourse); (2) code generation (the systematic coding of relevant and important data); (3) theme identification (the grouping of selected codes to form potential themes); (4) the continuous review of themes and identification of possible new themes; (5) theme definition, relating to the analysis and refinement of the specifics of each theme; and (6) the production of the final report (self-explanatory interpretation with the aggregation of data and themes) [17].

A rigorous reading of the answers to the questions was carried out, to allow the formulation of initial ideas, and an inductive thematic analysis was carried out, which aligns with the objective of this investigation. In the inductive thematic analysis, the data play a leading role, and the researcher’s conceptions should not be valued [17].

The central topics of the investigation include the identification of people with mental disorders, professional experiences with people with mental disorders, and coexistence with people with mental disorders. The answers to the questions were listed separately, and coding was carried out. Subsequently, the coded material was read again to group it into themes and subthemes, considering the relevance and coherence of the selected excerpts to list them and proceed with the refinement of the themes, which took place through reading the extracts, with a view of outlining a pattern for the central ideas, based on the objective of this investigation.

### 2.5. Ethical Aspects

This research was approved by the Ethics Committee on Human Research of the Municipal Health Department and the Nursing School of Ribeirão Preto at University of São Paulo (protocol number: 26431119.6.0000.5393). The Informed Consent Form was signed after being read and thoroughly explaining the participants the study objectives, its risks, and benefits, and the guarantee of anonymity, secrecy, and privacy. The application of the data collection instruments occurred in a reserved environment, considering the physical structure of each Health Unit, preserving the participant’s privacy as much as possible. To guarantee anonymity, we used acronyms followed by the numbers 1, 2, 3, and so on, to identify the professional categories, as shown in Table 1.

## 3. Results

After the analysis of the interviews, two themes and five subthemes were listed, as shown in Table 2.

### 3.1. From Personal Beliefs to Professional Experiences: Understanding People with Mental Disorders

Results show that the view regarding people with mental disorders is confused and mixed with the meaning given to mental disorders. In this sense, two subthemes were built, i.e., (1) interpreting concepts, contexts, and behaviors and (2) living together, deconstructing, and reinterpreting.

#### 3.1.1. Subtheme 1: Interpreting Concepts, Contexts, and Behaviors

Participants’ understanding of mental disorders reflected a blend of biomedical and lay perspectives. Mental disorders were often described as “invisible diseases” linked to brain function and typically identified through medical diagnoses. However, the role of medication was not viewed as universally essential for treatment.

The first thing that comes to my mind is an illness. […] In some cases, it doesn’t necessarily need drug treatment, but I always think of an illness related to the psychiatric part.(FHP 6)

It’s… an invisible disease.(CHA 27)

By the very part, that it affects the brain because they are diseases that affect the brain.(CHA 31)

Mental disorders that have already been evaluated by a doctor, that’s why the doctor evaluates them.(CHA 29)

It was also possible to identify that “anxiety crisis”, “bipolar disorder”, and “depression” were used sometimes as medical nomenclature and sometimes as common expressions. However, at the same time, the interest and the difficulty of professionals of the Family Health Care teams (FHTs) in differentiating symptoms that may or may not be characterized as a mental disorder, such as sadness and depression, are noticeable.

People who have anxiety crisis, fears, sadness, depression, you know.(CHA 3)

Well, generally, people who have some compromise with the psychic part, or are going through anxiety, depression or some kind of personality disorder.(PHARM 10)

People that… not only have mental health issues, but people with depression, sometimes it’s… in my case that I know, my cousin, he has schizophrenia, so, right from the start it comes first from mental health issues, but not only, also people with depression […] that patient who is always complaining… complaining… complaining… and sometimes I think it’s more of a complaint.(OHA 1)

Another important aspect is related to the association of people with mental disorders to a diagnosis with their respective symptoms.

Ah, I always associate that, and I tell the patients themselves this, that the difference between those who take medication and those who don’t is the intensity. So, the exacerbation, right, and everyday situations. So, anxiety, depression, insomnia, crazy outbursts, punching the wall, everyone has them. Now, the degree of this will differentiate the one I characterize as sick from the one I characterize as sane.(FHP 7)

People with mental disorders? Ah, there are several types of mental disorders, it varies a lot, a depression, a person who needs hospitalization, I think there are several stages of mental disorder, but I want…some people who see a mental disorder think that the person is…is totally unbalanced, I don’t see it that way, I see that there are several stages, from depression to the most acute state, the most serious state.(NUR TECH 1)

Additionally, there was a prevalent association of mental disorders with reduced autonomy and communication challenges. This perception perpetuated stereotypes about dependency and social dysfunction.

Let me find a word… they don’t take care of themselves. They are people who depend on others, most of the time. (CHA 2)

They have some psychiatric problem, it’s… sometimes they don’t have the same facility as other people to behave, for example, to come to the pharmacy and do a simple process that is to come and get a medication, they can’t have an effective medication, and we also have difficulty in communicating with them.(PHARM 2)

For some participants, people with mental disorders have a life marked by suffering, whose predominant feeling is sadness, the thought is meaninglessness, and the feeling is a lack of pep and mood.

Ah, I think it always goes through a little of that stereotype, right, I think the person starts to regret, always someone unhappy, someone with a difficult life, something discouraging, so, it’s what first passes through my mind. (CHA 1)

[…] They are people who suffer. They suffer since… since always. Furthermore, they suffer in the family, they suffer at work, they suffer… they suffer all their life. (NUR ASS 3)

It’s a person who has serious headaches and when they don’t take medicine, they are unfortunate, very discouraged. A person who sometimes gives up on life. (CHA 36)

The tension between recognizing these individuals’ challenges and avoiding stigmatizing descriptions was evident, reflecting participants’ struggle to articulate their perspectives without reinforcing binary notions of normality versus abnormality.

My view of a person who has a mental health problem, is that something in him stands out… and not because of something like, look how she’s working, she’s trying hard, or she’s studying. Even if you don’t know it… because I’m not a nurse, I’m not a psychiatrist, I’m not a doctor… but you look at it and you say, “there’s something”. (N 13)

Yes… when we say “disorder” it means that something is not right. (PHARM 3)

In this perspective, there was even difficulty for the professionals to explain these differences without running the risk of classifying people with mental disorders within the scope of normality x abnormality.

The anxious person is not that, he/she is, not that he/she is not normal, not that he/she is always… it is I don’t know how to explain it, it’s hard to explain, that something is out of the ordinary. (FHP 3)

We always imagine a person who is, not uncontrolled, but a person who is, who has a disorder, when you say mental disorder it’s because it’s a person that escapes, not from normality, that escapes from what is common for us […] When a person is under control, I understand that the illness is under control, you don’t notice the difference between a person who acts normally and another that doesn’t. (N 6)

For the participants, many people do not seek help, do not want to be helped, do not seek treatment, and do not accept that they have problems.

I see many people that, sometimes, I even think they have something, and the person doesn’t want to know, you know? They don’t want to be treated, they don’t want to find out… because of these characteristics, the person is always very depressed, always very unhappy.(CHA 13)

[…] Medication and treatment, and to ask for help. So, these people are the ones that are more difficult to deal with, for us to become aware that they need to ask for help.(CHA 34)

Thus, the beliefs of participants are influenced by the social imaginary, in which professional experiences based on a biomedical view of mental disorders are mixed with common sense and extended to a view of emotional suffering.

#### 3.1.2. Subtheme 2: Living Together, Deconstructing, and Reinterpreting

Hoping to overcome the negative beliefs, the participants shared reports about the idea built by the influence of the institutionalized model of mental health care, in which people were socially segregated, and their attitudes associated with the ideas of dangerousness and unpredictability.

[…] I used to see these people, you know, like… yeah… lying around, talking to each other, sitting down, screaming, and since I was a kid, I kept that in my mind, you know? The person who has a mental disorder is confined to a place and doesn’t have any ability to decide for himself because everything he does is supported by the place. And I remember when a patient came out of there that was going to do something, everyone used to say, ”don’t get too close because he’s a patient from the sanatorium, so you don’t know what he can do to you”. (N 1)

As time went by, when we worked in the health area, we realized that problems are not only related to madness, like in the old days, they are… so, I think that because I think that any kind of problem that disturbs your mental state is a disorder. So… I interpret… this is what helps me to think this way because I believe that the disorder is not only the one who is throwing stones (laughs). It’s anything that shakes your mental state. (PHARM ASS 3)

However, perceptions that express the re-signification of professional experiences are influenced by everyday experiences.

At first, it’s that prejudice, that you don’t know what’s going on… you already have a kind of labeling… but then with the involvement, with the passing of the consultations, there is an improvement… there is a different look that also depends a little on the professional. (CHA 11)

Yes… the work helped me to see differently, to see the history of the person, what he/she have experienced in that situation. (CHA 33)

These shifts were often tied to specific interactions with patients and their families, as well as efforts to challenge the stigma surrounding mental health. Participants highlighted the importance of demystifying mental disorders and promoting acceptance within the community.

I think it is significant that we demystify mental disorders, especially for the patient… because there are many young patients who come, their family says it is a lack of God, that they need to go to church… we must explain that it is not that… we don’t wake up and say “ah, today I want to have depression, today God is not with me”. (FHP 4)

Although personal and professional experiences contributed to a more empathetic understanding, participants acknowledged that deeply ingrained societal beliefs about mental disorders remain a significant barrier to comprehensive care.

### 3.2. Living with Mental Disorder: From Social Impacts to Potentialities

When reporting how they imagine living with a mental disorder, the participants revealed that the limitations arising from mental disorders do not determine the inability of these people to live socially, but they emphasized that there are negative social impacts and discrete opportunities.

In this sense, three subthemes were listed, i.e., (1) experiences and negative feelings: a life of oscillations and insecurity, (2) fragility in interpersonal relationships: stereotypes, social distance, and discrimination, and (3) needs, possibilities, and opportunities in the life of people with mental disorders.

#### 3.2.1. Subtheme 1: Experiences and Negative Feelings: A Life of Oscillations and Insecurity

Participants described the life of people with mental disorders as marked by a lack of control over thoughts, attitudes, and behaviors, which lead to insecurity in social relationships, fear, and feelings of worthlessness. These feelings of instability and vulnerability create a cycle of emotional suffering.

The emotional suffering that emerges from these disorders was also represented by accounts of loss of life perspective and references to suicide.

[…] I think that they don’t know that certain attitudes and certain ways that they transmit something, will sometimes harm someone, or sometimes may harm the person in some way, or make the other person upset… they should be afraid of having ups and downs, of having relapses, of the medication sometimes not working. Sometimes, they want to live together and being afraid of doing something to family, or friends, of people having prejudice against the person. (N 1)

I think that they get frustrated, sometimes afraid, sometimes angry, and sometimes irritated. (CHA 3)

The emotional suffering that emerges from these disorders was also represented by accounts of loss of life perspective and references to suicide.

The person, many times, keeps quiet, so you don’t know what goes through their head. Sometimes, they don’t open up because they don’t feel like talking. Then, when you look again, the person has already hurt themselves, cut themselves, or taken medication on their own because they want to end their lives. So, it is unfortunate. (CHA 5)

[…] I think that they feel terrible. Depending on the disorder, they feel awful. There are some that don’t even have the will to live. It comes to this point. (CHA 28)

[…] ah, it’s bad, you…the climate is heavy, right, you realize that it is heavy, that sometimes you talk to the person about everything, and the person has no… sees no way out. (N 4)

Also, participants reported that most people who have a mental disorder have traumas and sufferings that are frightening, when this situation is shared in the sessions.

[…] most of them have some trauma, so you see that the person is suffering, that’s why sometimes we get worried, sometimes the person’s speech is very shocking, in the sense that the person says how much it hurts, I think that’s more like it because they feel, I think they feel discrimination more than we do. (CHA 20)

The emotional burden of mental disorders, according to participants, is often aggravated by the unpredictability of symptoms, creating feelings of fear and helplessness. These accounts highlight how the stigma and self-perception of “lack of control” contribute to reinforcing feelings of vulnerability and isolation.

#### 3.2.2. Subtheme 2: Weaknesses in Interpersonal Relationships: Stereotypes, Social Distance, and Discrimination

The limitations imposed by mental disorders on people’s lives have been associated with social losses, and this leads to low social contribution, thus strengthening the feeling of worthlessness and powerlessness in the person with a mental disorder.

[…] a person who is active in everyday life and suddenly acquires a disorder must feel powerless… incapable of doing the things they used to do before… dependent on the help of doctors, medication, health professionals. I think that when this person develops a disorder, they feel like they’re a bit outside of society. (N 2)

The difficulties in relating, expressing, and actively contributing to the context in which they are inserted lead to social isolation, since the support network does not offer the emotional support necessary to deal with the issues related to suffering.

Ah, I imagine people with relationship difficulties, people who have difficulty with life. (NUR TECH 40)

The lack of social support and understanding often contributes to this sense of exclusion.

[…] A student I attend to has schizophrenia, and socially, people can’t understand what it is… They can’t give her the support she needs, and she automatically isolates herself. (FHP 1)

Thus, the lack of knowledge about the disorders and the prejudice in social interaction tend to generate a lack of understanding both in the person with the disorder and in the people who live with him/her, which causes damage in social relationships.

[…] Behavioral changes, especially in those with anxiety or panic disorders, are not very well understood. That’s why they suffer prejudice. People see them talking normally and, suddenly, they’re scared, sweating, their heart is racing… The other person tells them to calm down, but it’s not that simple. (CHA 23)

I think it is extremely crucial that we demystify mental disorders, especially for the patient… because there are many young patients who come, their family says that it is a lack of God, that they need to go to church… we have to explain that it is not like that… we don’t wake up and say ’ah, today I want to have depression, today God is not with me’. (FHP 4)

This reality creates barriers to accessing treatment, since people avoid sharing their feelings, thus reinforcing the self-stigma of people with mental disorders.

Many people who undergo treatment hide it even from their family members. This creates a barrier to treatment adherence, as dealing openly with the condition often makes the treatment more effective. (FHP 5)

It is noteworthy that the complexity that involves mental disorder leads to difficulties in relation to work/employment because many employers do not understand and even question the merit of certificates that ask for leave of absence due to psychiatric conditions.

[…] her employer came to the unit questioning why a doctor, after just talking to the patient, could give a seven-day certificate for mental illness. He said, ‘She hurt her knee before, and that’s understandable. But this? Seven days off for these things?” (N 2)

In addition, the loss of essential skills for the performance of work and daily activities (studying and socializing) causes a feeling of incapacity to seek a new work activity.

In fact, it is a loss of ability, isn’t it? The first thing that one notices is that the patient has difficulty at work, at studying, in interpersonal relationships, so the patient with a mental disorder is really characterized, it’s a patient who needs medication to be able to return to a normal life. (FHP 7)

I think that some difficulties that we see, difficulties in solving problems, people find it difficult, or to get a job, they believe that they are incapable, that they don’t have the capacity, you know, in that sense. (N 4)

Above all these difficulties, participants report that people with mental disorders also face barriers in their family contexts due to the lack of support from family members who do not understand their condition.

[…] I think it is very difficult, individually, for themselves right, for all the difficulty that this brings and for the family because the family can’t understand, sometimes they think that depression is cool, silly, and it’s not like that, we understand and know they need treatment. (CHA 4)

[…] avoid contact with other people, and so, this is present even among them right, among families, sometimes family members exclude these people right, they don’t make a point of visiting, they ignore, so they suffer, it’s… as a whole. (N 3)

Thus, the idea that people with mental disorders damage their social relations was present in the speeches of some of the professionals interviewed.

Ah, very complicated because it is not normal for a person to live like this, so I think it must be so, not only for the person, but for the family members too, it must be very difficult to have a person with a mental disorder at home. (PHARM ASS 1)

Horrible, it affects everything, everyone looks desperate and doesn’t know what to do, especially close relatives, for me, it is desperate, you have a volcano exploding next to you at any moment, even with less serious problems, it is desperate, a milder bipolar disorder is bearable […] A person with a mental disorder harms many people around him, his mental problem harms many people, causes many problems, that is why, I think that he must be treated, he can’t leave it alone. (CHA 6)

Such social relationships are hampered by difficulty in following rules and social isolation, according to the participants.

They are a little reduced because they can’t always understand the right way to do things, the rules that they have, right? I think that’s it… I think it’s limited. (CHA 22)

Ah… they don’t feel well. They feel isolated, they are afraid of relationships, afraid of making friends. They feel lonely… in my opinion, they are lonely… sometimes you think not, but they suffer. (CHA 39)

In this way, professionals show that the person needs to deal not only with issues pertaining to mental suffering but also with discrimination and feelings of inferiority, which begin to circumscribe family and social relationships.

It is a difficult life because in addition to having to deal with the disease itself, she must deal with discrimination. (DEN 5)

[…] I think that is it, in general, there are people who deal with it very well, but in the area that we are here, most of them don’t have this access, they don’t like it very much, they feel inferior. (FHP 3)

These accounts underscore how mental health conditions disrupt social and work relationships and how stigma and ignorance contribute to self-imposed and socially-imposed isolation.

#### 3.2.3. Subtheme 3: Needs, Possibilities, and Opportunities in the Lives of People with Mental Disorder

Treatment is highlighted as a crucial point for the improvement and stabilization of the mental disorder, and in this context, medication stands out.

If treated properly… People who adhere to treatment, who take their medication correctly, can work and maintain family relationships. Even though there are setbacks, they seek help and recover. (CHA 30)

Certainly, even if it is mild, it ends up hurting a little. But we know that with medication the person can continue the routine.(CHA 31)

In fact, it is a loss of ability, right? The first thing that one notices is that the patient has harm at work, at study, in interpersonal relationships, so the patient with mental disorder is really characterized, it’s a patient who needs medication to be able to return to a normal life. (FHP 7)

It has been reported that a positive characteristic present in people with mental disorders is intelligence, especially among those with more severe conditions, but it is little exploited due to lack of opportunities in life. Yes. There are people I think, even if you go deeper, that they even have an IQ a little higher than the population, right. And I think that I also feel that they are people who feel more, have more sensitivities, right? (PED 1)

It was pointed out that mental disorders do not lead to disability, but that there can be the depreciation of abilities of the user if he “sees” himself as someone sick and limited.

Most of them do, right, sometimes with a certain difficulty, but they can do it. (NUR TECH 4)

They all have the same capacity as a person who does not have a mental disorder. Maybe they don’t have the initiative… nor the initiative that I would say… because of the mental disorder they think they are incapable of. But he has the capacity of a person without disorder.(CHA 28)

I believe that they have abilities as ordinary people…they are ordinary people, really, they are just going through a difficult time. But I believe that they can have any ability like a person who is not suffering from a mental illness has as well.(CHA 31)

For the participants, the needs, possibilities, and opportunities will also depend on the degree or level of their mental disorder.

[…] it depends on the degree, right, from individual to individual, I believe it is particular to each one, it is a spectrum, right, that has several levels, so there are people who have some minor limitations, others greater, but you can’t generalize, it depends on each one, at least what I see here… (PHARM 2)

I think it depends a lot on the degree of the disease that the person is living now. I think that most of them manage to have a normal routine, but there are some cases, when it is in an acute phase, I think that… the person ends up losing his life, in the sense of not being able to do anything, not being able to live. (CHA 31)

Undoubtedly, it is necessary to see and develop more carefully the care and conditions to meet the needs of people with mental disorders, through their own resources and abilities, which requires working towards stigma-free practices.

## 4. Discussion

International studies [18,19] point out that there is a reduction in stigma among professionals with more knowledge about mental disorders. This reality corroborates the findings of this research, since it was identified that the participants have different definitions of people with mental disorders, which are predominantly based on the biomedical model with emphasis on the difficulty of professionals to deal with subjectivity and, consequently, in the devaluation of the assumptions of the Psychiatric Reform, which tends to lead to stigmatizing conceptions.

Despite the different conceptions about mental disorders and regarding the people affected by them, results showed an exercise of some participants to understand the history of suffering from these people and this may be related to the time of training of these professionals, since it was mentioned in a systematic review that younger and less experienced professionals who work in primary care are more sensitive to the reality of the person with a mental disorder, which makes them have less stigmatizing attitudes [19]. However, a recent study [20] points out that the lower attribution of stigma is related to greater knowledge of mental health, training in mental health, and the performance of activities in mental health.

Even so, it was identified in this research that there are negative conceptions that, according to the participants, result from the period before the Psychiatric Reform, when the institutionalization of people with mental disorders was prioritized and this was also discussed in scientific research [21] that pointed out the need to value the intersection between stigma and current policies.

However, there is a movement to re-signify these beliefs and practices, which is influenced by personal beliefs, by previous social experiences, and by experiences of professional practice, which seem to be enabling the construction of a perception that these people experience suffering and negative impacts due to the illness. In this sense, investment in educational practices and the use of matrix support to empower professionals for this process of re-signification are paramount, since it is necessary to strengthen the psychosocial model that enables the remission of stigmatizing definitions and behaviors [18,20,22,23].

Negative social impacts represented by emotional suffering appear in the speeches of the participants, who perceive that insecurity and unpredictability permeate the life of a person with a mental disorder. One of the reasons for the barrier of access of people with mental disorders to primary health care (PHC) is this labeling of unpredictability that brings stigma and consequently a deficiency in the actions in mental health. It is believed that this reality is accentuated by the findings of this investigation, in which the participants mention that there is fragility in interpersonal relationships arising from social isolation and self-stigma of people with mental disorders that leads to denial of the need for treatment [23].

Moreover, it was evidenced in this study that the lack of knowledge of family members about the pathology generates stereotypes, social distance, and discrimination, which seem to accentuate the difficulties that already exist for PHC professionals to establish a link with people with mental disorders [23]. In general, the family nucleus is affected by the presence of a member with a mental disorder, which is expressed by emotional and financial weaknesses, triggering overload [24].

This reality reveals the importance of the support network in the lives of people with mental disorders, in which the family is key to treatment adherence. In this context, the existence of stigma represents a barrier to treatment and other possibilities for improving the quality of life [25] and predisposes to the exacerbation of mental disorders, greatly compromising the prognosis [26].

Health professionals report insecurity and doubts about their ability to provide effective treatment, which is accentuated by the presence of family members during the therapeutic process [27]. In this aspect, in the perception of the participants of this research, treatment is essential for the person to recognize himself/herself as having a mental disorder, control the symptoms, and, despite the imposed limitations, achieve a quality of life.

It was also mentioned in this research that it is important to consider the particularities of each person, and some will have greater and others lesser limitations, since the spectrum that covers the disorders is varied and complex, but even in the face of these difficulties, people in pain do not have any disadvantage in terms of their abilities and capabilities. This finding seems to represent a stigma, since it is denied that some severe mental disorders (for example, schizophrenia spectrum, autistic spectrum, etc.) can bring about limitations in the abilities and capacities of people, which needs to be considered in the provision of primary health care and does not ensure the withdrawal or restriction of access to fundamental rights, especially the right to health [28,29]. In this sense, the Brazilian scenario highlights that equity is one of the doctrinal principles of the Unified Health System, in which the “difference” needs to be valued to qualify and individualize health care [9].

From this perspective, professional support was pointed out by the participants as essential for people with mental disorders to have the opportunity to give new meaning to their lives. To this end, it is necessary to structure continuing education programs in primary care, since the recent research developed in India [30] detected a significant improvement in the knowledge and attitudes of nurses who received in-service training on the topic of mental health.

Even with the limitation of being developed in only six Family Health Units of a city in the interior of São Paulo, Brazil, the findings of this study express the need to invest in interventions to eliminate the existing stigma among PHC professionals, through educational and awareness-raising actions, since this level of health care plays a leading role in the organization of the Brazilian Health System. Another limitation to be considered is the joint analysis of health professionals, without the distinction of category. To fill this gap, future studies that comparatively investigate the specific perceptions of each professional category are imperative.

## 5. Conclusions

The results of this study demonstrate several stereotypes that reinforce what is considered normal and abnormal by health care professionals working at FHUs. Therefore, stigma is present in the perceptions of PHC professionals, who use biomedical conceptions to define people with mental disorders. This reality is also present when the challenges in the lives of people with mental disorders were pointed out, among which were self-stigma, social isolation, and difficulty in interpersonal relationships.

In this complex context, although participants consider treatment crucial for the improvement and stabilization of mental disorders, they highlight barriers faced by persons with mental disorders to access treatment and to live a life with dignity.

Furthermore, the results of this research indicate that, among insecurity, unpredictability, fear, emptiness, confused emotions, perceptions of helplessness and uselessness, loss of status and even discrimination, people with mental disorders may find, in health professionals, the possibility of some transformation. In order for this to be possible, it is necessary to foster initiatives involving health care professionals at UFHs to deconstruct beliefs that reinforce stigma and cause even more social losses. Undoubtedly, providing legitimate primary health care to people with mental disorders is essential to assure the continuity of their treatment, increasing future possibilities of rehabilitation and recovery.

## Figures and Tables

**Table 1 healthcare-13-00243-t001:** Professional categories and related acronyms.

Professional Category	Acronym
Nurse	N
Nursing assistant	NUR ASS
Nursing technician	NUR TECH
Family health physician	FHP
Pediatrician	PED
Pharmacist	PHARM
Pharmacist assistant	PHARM ASS
Dentist	DEN
Oral health assistant	OHA
Community health agent	CHA

**Table 2 healthcare-13-00243-t002:** Thematic themes and subthemes identified in the interviews’ content.

Themes	Subthemes
From personal beliefs to professional experiences: understanding people with mental disorders	Interpreting concepts, contexts, and behaviors
	Coexisting, deconstructing, and reinterpreting
Living with mental disorder: from social impacts to potentialities	Negative experiences and feelings: a life of oscillations and insecurity
	Fragility in interpersonal relationships: stereotypes, social distance, and discrimination
	Needs, possibilities, and opportunities in the lives of people with mental disorders

## Data Availability

The raw data supporting the conclusions of this article will be made available by the authors on request.

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
