# Peer review of "Perceptions of Family Health Strategy Professionals About People with Mental Disorders"

_healthcare, 2025, doi:10.3390/healthcare13030243_

Round 1

Reviewer 1 Report

Comments and Suggestions for Authors

Dear Authors,

Thank you for providing me with the opportunity to read this manuscript.  Below I have listed my comments:

1) In the abstract, it would have been much better to state some of your findings rather than simply mention the titles of the themes/sub-themes.

2) In the methods, line 16, it needs to be 'was carried out through individual'.

3) Try to expand slightly on barriers like prejudice among healthcare professionals. Why is this a significant problem, and how does it impact care delivery? Providing a specific example or relevant studies would strengthen this section.

4) It would have been nice to also add some relevant information about professionals' perceptions regarding people with mental disorders. There are some studies in Brazil such as 

Ghesquiere, A. R., Pinto, R. M., Rahman, R., & Spector, A. Y. (2015). Factors Associated with Providers' Perceptions of Mental Health Care in Santa Luzia's Family Health Strategy, Brazil. International journal of environmental research and public health13(1), ijerph13010033. https://doi.org/10.3390/ijerph13010033

5) In the methods, the explanation of the thematic analysis is adequate but could include examples of themes or codes identified to illustrate the process. Rather than stating the steps, it would have been better to explain the phases in a more detailed manner. Did you use indictive or deductive coding?

6) How did you conclude to these themes? How did you agree on them? What happened if there was a disagreement? How did you preserve objectivity? These are questions that need to be addressed.

7) Ensure consistent use of terms like 'participants', 'professionals', or 'FHS team members' throughout the section.

8) In the findings, it is stated that there were 82 participants- is this correct? did you conduct 82 interviews? I think that the number of participants would have been better to be stated in the participants sub-section so that the reader is aware of the sample before reading the findings.

9) For the findings, try to maintain a balance between interpretations and quotes. Now quotes are lengthy and there is a brif interpretation. It would have bene better to explain the quotes and provide a more latent analysis of their meaning.

10) The discussion briefly mentions the role of families but does not fully explore their importance or challenges.

11) The findings around stigma and biases can be expanded further by looking into other studies too which have found biases in the treatment of mental disorders. This way, you can highlight the consequences such biases can have globally on people with mental disorders. I have listed one below which I read recently about anorexia

Tragantzopoulou, P., & Giannouli, V. (2023). “You feel that you are stepping into a different world”: Vulnerability and biases in the treatment of anorexia nervosa. European Journal of Psychotherapy & Counselling, 25(4), 351–368.

12) The recommendation for continuing education is noted but not detailed. Can ypu provide more explicit suggestions?

I hope this feedback is helpful.

Author Response

Comments 1: In the abstract, it would have been much better to state some of your findings rather than simply mention the titles of the themes/sub-themes.

Response 1: Thanks for pointing this out. We agree with this comment. Therefore, in the summary results, we present the main results based on the themes and subthemes listed [page 1, line 20 to 22] Findings showed the existence of a concept of people with mental disorders based on the biomedical paradigm and that they experience limitations arising from the disease that cause restrictions of opportunities, even though they have the capacity to live socially.

Comments 2: In the methods, line 16, it needs to be 'was carried out through individual'.

Response 2: We agree. We set it to "was carried out through individual..." [page 1, line 17]

Comments 3: Try to expand slightly on barriers like prejudice among healthcare professionals. Why is this a significant problem, and how does it impact care delivery? Providing a specific example or relevant studies would strengthen this section.

Response 3: We agree. We expand on some of these barriers and answer the question [page 2, line 80 to 84] This reality is expressed in the fact that FSH's multidisciplinary mental health care focuses on referrals to specialized services, which represents a reflection of the prejudice of these professionals who view all people with mental disorders as beings who are not capable of monitoring at this level.

Comments 4: It would have been nice to also add some relevant information about professionals' perceptions regarding people with mental disorders. There are some studies in Brazil such as Ghesquiere, A. R., Pinto, R. M., Rahman, R., & Spector, A. Y. (2015). Factors Associated with Providers' Perceptions of Mental Health Care in Santa Luzia's Family Health Strategy, Brazil. International journal of environmental research and public health, 13(1), ijerph13010033. https://doi.org/10.3390/ijerph13010033

Response 4: We agree. So, we included some of these insights based on the suggested study, which was duly referenced. [page 2, line 84 to 86]. “It is also noteworthy that in the Brazilian organizational system, community health agents have been providing more care centered on patients with mental disorders, when compared to doctors and nurses who work at FSH [15]”

Comments 5: In the methods, the explanation of the thematic analysis is adequate but could include examples of themes or codes identified to illustrate the process. Rather than stating the steps, it would have been better to explain the phases in a more detailed manner. Did you use indictive or deductive coding?

Response 5: We agree. Therefore, we include a more detailed explanation of the inductive thematic analysis process, including the codes. [page 4, line 153 to 160] “A rigorous reading of the answers to the questions was carried out, to allow the formulation of initial ideas and an inductive thematic analysis was carried out, which aligns with the objective of this investigation. In inductive thematic analysis, the data plays a leading role and the researcher's conceptions should not be valued [17]. Central topics of the investigation – identity of people with mental disorders, professional experiences with people with mental disorders and coexistence with people with mental disorders. The answers to the questions were listed separately, and coding was carried out.

Comments 6: How did you conclude to these themes? How did you agree on them? What happened if there was a disagreement? How did you preserve objectivity? These are questions that need to be addressed.

Response 6: These questions are extremely important. They have been answered. [page 4, line 160 to 164] ”Subsequently, the coded material was read again to group it into themes and subthemes, considering the relevance and coherence of the selected excerpts to list them and proceed with the refinement of the themes which took place through reading the extracts. , with a view to outlining a pattern for the central ideas, based on the objective of this investigation.

Comments 7: Ensure consistent use of terms like 'participants', 'professionals', or 'FHS team members' throughout the section.

Response 7: We chose to leave the term "participants" throughout the text, although in some contexts the other two terms become indispensable. [page 7, line 293 and 298; page 8, line 344; page 9, line 376; page 10, line 444]

Comments 8: In the findings, it is stated that there were 82 participants- is this correct? did you conduct 82 interviews? I think that the number of participants would have been better to be stated in the participants sub-section so that the reader is aware of the sample before reading the findings.

Response 8: Yes. There were 82 participants interviewed by the research team described in the method section. This number reflects the large number of professionals in health units and the different perspectives and experiences that the study aimed to understand. We agreed to explain about the participants in the subsection. That's why we rearranged this information. [page 3, line 109 to 101] “A total of 82 professionals participated in the research, 82.9% were female, and most of them were between 36 and 57 years old (58.9%). More than half declared they had completed college education (59.8%). Regarding education, there was a prevalence of middle level courses, such as nursing assistant (3.7%), nursing technician (7.3%), oral health assistant (4.9%) and courses like radiology, HR management and others (12.2%). Still, there was a prevalence of training in higher education courses that do not include those who integrate the Family Health Team (ESF) (38.9%). This is due to Community Health Agents (CHAs) and also to those professionals who have other educations besides the one performed in the FHU. According to the 'professionals' statements, the following occupations predominated: 53.7% CHAs, 8.5% family health doctors, 6.1% nurses, 6.1% dentists, and 6.1 oral health assistants. The majority (42.5%) declared they had been working at the FHU between one and five years, and 8.5% between 16 and 20 years.

Comments 9: For the findings, try to maintain a balance between interpretations and quotes. Now quotes are lengthy and there is a brif interpretation. It would have bene better to explain the quotes and provide a more latent analysis of their meaning.

Response 9: We thank you for your evaluation and acknowledge the need to maintain an appropriate balance between participants' restrictions and the interpretations made. We understand that the inclusion of extensive excerpts without a more in-depth analysis can make the presentation of results descriptive and limit the identification of the latent meanings of the statements. Therefore, we reduced the extent of the restrictions, deepened the analysis, balanced both. The adjustments are in red in the results section.

Comments 10: The discussion briefly mentions the role of families but does not fully explore their importance or challenges.

Response 10: We agree. We complement these aspects in the discussion. [page 13, line 561 to 563] “In general, the family nucleus is affected by the presence of a member with a mental disorder, which is expressed by emotional and financial weaknesses, triggering overload [24].”

Comments 11: The findings around stigma and biases can be expanded further by looking into other studies too which have found biases in the treatment of mental disorders. This way, you can highlight the consequences such biases can have globally on people with mental disorders. I have listed one below which I read recently about anorexia: Tragantzopoulou, P., & Giannouli, V. (2023). “You feel that you are stepping into a different world”: Vulnerability and biases in the treatment of anorexia nervosa. European Journal of Psychotherapy & Counselling, 25(4), 351–368.

Response 11: We agree. We complement this part of the discussion according to the suggested reference. [page 13, line 569 to 571] “Health professionals report insecurity and doubts about their ability to provide effective treatment, which is accentuated by the presence of family members during the therapeutic process [27].”

Comments 12: The recommendation for continuing education is noted but not detailed. Can ypu provide more explicit suggestions?

Response 12: The concluding session was rewritten, in which more suggestions were detailed. [page 14, line 613 to 618]. “In order for this to be possible, it is necessary to foster initiatives involving healthcare professionals at UFHs to deconstruct believes that reiforce stigma and cause even more social losses. Undoubtedly, providing a legitimate primary healthcare to people with mental disorders is essential to assure the continuity of the treatment, increasing future possibilies of rehabilitation and recovery.”

Reviewer 2 Report

Comments and Suggestions for Authors

The  problem of medical and mental health professionals holding the same stigmatizing attitudes as the general public is huge. I hope the authors will continue to work in this area and fine-tune their work.

The title of the article is an indicator of other problems, especially regarding issues of translation. "Strategy professionals" doesn't seem to be a correct translation. I think the article is more appropriately titled: Perceptions of the mentally ill by medical professionals in Brazil."

The literature review needs to give a global picture of types of  mental illness and the percentages of individuals who suffer from each type.

The most important criticism is that the authors need to distinguish between types of mental illness. Individuals with anxiety disorders and depression are not likely to be as obviously mentally ill as those with schizophrenia spectrum disorders. Placing the range of mental disorders into one category is simply not sufficient. Rather, it can add to the problem of stigma by placing all individuals with mental illness, regardless of diagnosis, as worthy of judgment and harsh labeling.

The assumption seems to be that the professionals/workers in the field, regardless of specific job category, will hold the same attitudes. That is not necessarily the case.

The sample size of 82 medical respondents is a very good sample size for qualitative data.

Asking "who are people with mental disorders?" is simply too broad. The first theme is not clear; the  second theme, with the subthemes, is better. The authors need to do more more to craft better questions for respondents.

The Results section reads like a laundry list of quotes from respondents. It is merely descriptive and not analytical. While the Discussion section is much better than the Results section, it still needs to more clearly delineate the major concerns and findings.

The  Conclusions section is weak--it can be improved by offering more  specific recommendations for researchers and medical professionals as  well.

Comments on the Quality of English Language

The quality of English language is not strong. I suggest the authors work with a native speaker of English on a revised draft.

Author Response

Comments 1: The title of the article is an indicator of other problems, especially regarding issues of translation. "Strategy professionals" doesn't seem to be a correct translation. I think the article is more appropriately titled: Perceptions of the mentally ill by medical professionals in Brazil."

Response 1: Regarding the title, the Family Health Strategy are teams that work in primary health care in Brazil, as explained in the introduction [page 2, line 53 to 59]. Furthermore, the participants were not only doctors, but also nurses, pharmacists, among others. We understand that the translation may seem confusing at first, but our suggestion is that the title remains as it is, considering the research problem, the scenario and the audience studied.

Comments 2: The literature review needs to give a global picture of types of  mental illness and the percentages of individuals who suffer from each type.

Response 2: We agree and understand the importance of this panorama. Therefore, in the first paragraph of the introduction we provide data on the common mental disorders most diagnosed worldwide and in Brazil, such as anxiety and depression. In relation to serious mental disorders, we focus on schizophrenia." [page 1, line 29 to 36]

Comments 3: The most important criticism is that the authors need to distinguish between types of mental illness. Individuals with anxiety disorders and depression are not likely to be as obviously mentally ill as those with schizophrenia spectrum disorders. Placing the range of mental disorders into one category is simply not sufficient. Rather, it can add to the problem of stigma by placing all individuals with mental illness, regardless of diagnosis, as worthy of judgment and harsh labeling.

Response 3: We thank you for your comment and recognize the importance of distinguishing between different types of mental disorders, as each condition has unique characteristics that influence not only the symptoms and behaviors observed, but also the way individuals are perceived socially. We agree that grouping all disorders into a single category can reinforce stereotypes and contribute to generalizations that do not reflect the diversity of experiences lived by people with different mental disorders. However, we would like to highlight that the focus of this study was to understand participants' general perceptions about the experiences of people with mental disorders, regardless of the specific diagnosis. The objective was not to detail the impacts related to each disorder, but to explore how interviewees perceive the difficulties and potential of people with mental disorders in a broad sense. Therefore, we reformulated several excerpts in the results and discussion section to make it clear that the participants' statements reflect their perceptions of different conditions, without implying that all experiences are identical. [page 5 - line 182, 184, 192 to 195; page 6 - line 232 to 234; 253 to 255; page 7 - line 287 to 290; page 8 - line 312 to 315, 321 to 323, 336 to 341, 351 to 352; page 9 - line 369 to 372, 388 to 389; page 11 - line 466 to 468; page 13 - line 557 to 559, 565 to 567]

Comments 4: The assumption seems to be that the professionals/workers in the field, regardless of specific job category, will hold the same attitudes. That is not necessarily the case.

Response 4: We thank you for your comment and recognize the relevance of your observation. In fact, health professionals may have different perceptions and attitudes towards people with mental disorders, depending on their professional category, training and experience. This diversity is an important aspect to be considered in studies on mental health. However, the objective of this study was to capture general perceptions of healthcare workers as a professional group, understanding that, despite possible variations, there are also shared perceptions that can be analyzed together. Still, we agree that a more detailed analysis by professional category could enrich the discussion and offer more specific contributions. Therefore, we included this issue in the limitations section, pointing out the need for future studies that comparatively investigate the specific perceptions of each professional category. [page 13, line 593 to 596]. “Another limitation to be considered is the joint analysis of health professionals, without distinction of category. To fill this gap, future studies that comparatively investigate the specific perceptions of each professional category are imperative.”

Comments 5: The sample size of 82 medical respondents is a very good sample size for qualitative data.

Response 5: We appreciate the recognition regarding the sample size used in this study. We agree that having 82 participants is a robust sample for qualitative research, especially in studies that seek to understand complex perceptions and experiences. It is important to highlight that the main objective was to guarantee a diversity of voices and professional categories, which contributed to a greater wealth of data and theoretical saturation. This sample allowed us to identify consistent patterns in participants' perceptions, in addition to understanding specific nuances related to the topic investigated.

Comments 6: Asking "who are people with mental disorders?" is simply too broad. The first theme is not clear; the  second theme, with the subthemes, is better. The authors need to do more more to craft better questions for respondents.

Response 6: We agree with the note, so we adjusted the name of the first theme to "From personal beliefs to professional experiences: understanding people with mental disorders” [page 5 - line 182 to 184]

Comments 7: The Results section reads like a laundry list of quotes from respondents. It is merely descriptive and not analytical. While the Discussion section is much better than the Results section, it still needs to more clearly delineate the major concerns and findings.

Response 7: We appreciate your feedback and confirm the importance of avoiding a merely descriptive presentation of qualitative data. We understand that the results section must go beyond reproducing excerpts from the participants' speeches, bringing an interpretative analysis that explains the latent meanings and the main findings of the study. To meet your suggestions, we restructured the results section, replacing extensive lists of specifications with more synthetic and representative excerpts. We expanded the interpretative analyzes after each quote, highlighting how the reports illustrate the main themes and subthemes. We clearly highlight the emerging and divergent patterns observed in the discourses. In the discussion section, we reinforce the connection between the research findings and the central questions of the study. We also add reflections on implications and recommendations based on the main concerns observed. The settings are in red.

Comments 8: The Conclusions section is weak--it can be improved by offering more  specific recommendations for researchers and medical professionals as well.

Response 8: We thank you for your observation and agree that the conclusions section could be improved to provide more specific and practical recommendations for both researchers and healthcare professionals. We adjusted the entire session. [page 13 - line 598] “The results of this study demonstrate several steriotypes which reinforce what is considered normal and abnormal by healthcare professionals working at FHUs. Therefore, stigma is present in the perceptions of PHC professionals, who use biomedical conceptions to define people with mental disorders. This reality is also present when the challenges in the lives of people with mental disorders were pointed out, among which were self-stigma, social isolation and difficulty in interpersonal relationships. In this complex context, although participants consider treatment crucial for the improvement and stabilization of the mental disorder, they highlight barriers faced by persons with mental disorders to access treatment and to live a life with dignity. Furthermore, the results of this research indicate that among insecurity, unpredictability, fear, emptiness, confused emotions, perceptions of helplessness and uselessness, loss of status and even discrimination, people with mental disorders may find, in health professionals, the possibility of some transformation. In order for this to be possible, it is necessary to foster initiatives involving healthcare professionals at UFHs to deconstruct believes that reiforce stigma and cause even more social losses. Undoubtedly, providing a legitimate primary healthcare to people with mental disorders is essential to assure the continuity of the treatment, increasing future possibilies of rehabilitation and recovery.”

Round 2

Reviewer 1 Report

Comments and Suggestions for Authors

Dear Authors,

Thank you for responding to my feedback. I am now satisfied with the current form of your paper. Good luck with the publication.

Author Response

Comment 1: Thank you for responding to my feedback. I am now satisfied with the current form of your paper. Good luck with the publication.

Response 1: Thank you very much for your kind words and for taking the time to review our manuscript. We sincerely appreciate your valuable feedback, which has greatly contributed to improving the quality of our work. We are delighted to know that you are satisfied with the current version of the paper. Thank you again for your thorough evaluation and support.

Reviewer 2 Report

Comments and Suggestions for Authors

I  recognize that the authors have  responded to some suggestions for strengthening their work.  I do not accept the explanation for why it's ok to  lump all mental disorders together and expect meaningful feedback from medical  professionals.

Author Response

Comment 1: I recognize that the authors have  responded to some suggestions for strengthening their work.  I do not accept the explanation for why it's ok to  lump all mental disorders together and expect meaningful feedback from medical  professionals.

Response 1: Thank you for your valuable feedback and for pointing out the limitations of grouping all mental disorders together in our analysis. We understand and respect your concern regarding the potential challenges this approach poses in obtaining meaningful and differentiated feedback from medical professionals.

Our decision to address mental disorders collectively was based on two main factors:

  1. Study Objective:
    The primary aim of our study was to explore general perceptions of Family Health Strategy (FHS) professionals about individuals with mental disorders. Our focus was on understanding broad patterns of stigma and barriers in care, rather than the clinical nuances of specific mental disorders. This approach reflects how these professionals often conceptualize mental health conditions in their practice, which tends to generalize mental disorders rather than differentiate between categories.

  2. Context of Brazil’s Healthcare System:
    In Brazil’s Primary Health Care (PHC) model, the FHS serves as the main entry point for a wide range of health conditions, including all types of mental disorders. Professionals within this setting are often required to provide care for both common and severe mental health conditions, and their perceptions naturally span across this spectrum. By taking a comprehensive approach, we were able to capture their overarching attitudes and challenges in delivering care, which are influenced by this broad scope of practice.

While we recognize the limitations of this approach, we believe that our study provides valuable insights into the general perceptions and challenges faced by FHS professionals when addressing mental health conditions. Future research may explore the perceptions of healthcare professionals regarding specific categories of mental disorders in greater depth, building on the foundation provided by our work.

We greatly appreciate your constructive critique and thoughtful engagement with our manuscript, which has helped us strengthen the discussion and rationale behind our methodological choices.